# Relationship between Medial Elbow Pain, Flexor Pronator Muscles, and the Ulnar Nerve in Baseball Players Using Ultrasonography

**DOI:** 10.3390/healthcare11010050

**Published:** 2022-12-24

**Authors:** Issei Noda, Shintarou Kudo, Kengo Kawanishi, Naoya Katayama

**Affiliations:** 1Graduate School of Health Sciences, Morinomiya University of Medical Sciences, Osaka 559-8611, Japan; 2Ashiya Orthopedics Sports Clinic, Ashiya 659-0092, Japan; 3Inclusive Medical Science Research Institute, Morinomiya University of Medical Sciences, Osaka 559-8611, Japan; 4AR-Ex Medical Research Center, Tokyo 158-0082, Japan; 5Department of Rehabilitation, Kano General Hospital, Osaka 531-0041, Japan; 6Osaka Gyoumeikan Hospital, Osaka 554-0012, Japan

**Keywords:** elbow, ulnar nerve, medial elbow pain, ultrasound, throwing injury

## Abstract

We aimed to clarify changes cross-sectional area (CSA) in flexor pronator muscles and the ulnar nerve (UN) in players with medial elbow pain between pitching phases. Forty-two male baseball players with and without medial elbow pain during throwing were included in this study. The players were divided into maximum external rotation (MER) and ball release (BR) groups according to the pitching phase in which pain was reported. The imaged region was the flexor digital profundus, flexor carpi ulnaris (FCU), flexor digitorum superficialis, and pronator teres muscles, as well as the UN. CSA at rest and during contraction was assessed using the ultrasonography software tracing function. For statistical analysis, the CSA at rest and at contraction in the healthy group, MER group and BR group was compared using one-way analysis of variance. There was a significant difference in CSA only in the FCU between the healthy (95.4 ± 15.5%) and the MER group (76.6 ± 12.5%) at rest (*p* = 0.004). There were significant differences in the UN between the healthy (105.0 ± 27.7%) and MER groups (176.4 ± 53.5%), and between the healthy and BR groups (132.9±21.1%) (*p* = 0.001 and *p* = 0.038, respectively). Our results suggest that athletes with medial elbow pain during the MER of pitching have ulnar nerve swelling.

## 1. Introduction

Medial elbow injuries are caused by excessive valgus torques at the elbow during baseball pitching [1]. The ulnar collateral ligament (UCL) is the most important static stabilizer of the elbow under valgus forces. In addition, structures other than the UCL may play a role to stabilize the elbow, such as the flexor pronator muscles and heads of the radius [2,3].

Repeated valgus load during pitching may lead to UCL dysfunction and valgus instability of the elbow. However, the relationship between the UCL injuries (dysfunction) and medial elbow pain with throwing are unclear. Ciccotti et al. assessed the UCL thickness, the gap of the ulnohumeral joint space with valgus stress, and echo-textural abnormalities (hypoechoic foci and calcifications) of the medial elbow, and reported no significant difference between a group of individuals with UCL injury and a healthy group [4]. In our previous study, we showed that baseball players with or without pain exhibit valgus instability [5]. Combined injuries of the flexor pronator muscles and UCL may occur among pitchers with valgus instability [6,7]. In addition, it has been reported that strains or tears may develop in the forearm flexors and affect the ability of a professional athlete to throw, resulting in significant time on the disabled list (DL) in Major League Baseball (MLB) [6]. Moreover, ulnar neuropathy is a common elbow injury in pitchers and is reported to be caused by compression of the ulnar nerve at the elbow [8,9]. Medial elbow pain has significant pathologies affecting the thrower’s elbow, including valgus extension overload, medial epicondylalgia, ulnar nerve pathology, and common the flexor pronator muscles injury [10,11]. Therefore, it is necessary to evaluating anatomic structures is to diagnose the cause of medial elbow pain. Abnormal findings of UCL are not always related to medial elbow pain, thus, the cause of pain may be in tissues other than the UCL, such as the flexor pronator muscles and UN.

Recently, medial elbow pain was assessed using ultrasound (US) imaging [10]. Regarding the flexor pronator muscles, it has been reported that the flexor carpi radialis (FCR) and the pronator teres (PT) muscles function as dynamic stabilization mechanisms against valgus stress [2]. However, that study was conducted in healthy adults and not in athletes with medial elbow pain. Measurement of the cross-sectional area (CSA) of the ulnar nerve have good sensitivity and specificity in diagnosing UNE [12]. Wiesler et al. reported that the UN with cubital tunnel syndrome had an increased CSA and that an increased CSA of the UN had a negative correlation with motor nerve conduction velocity [13]. Although reports on the flexor pronator muscles and UN have been published, few reports have described a relationship between the flexor pronator muscles and the UN.

A great load is applied to the medial elbow during the MER and ball release (BR) during throwing [14,15]. It has been reported that during the MER, shoulder rotational torque and elbow valgus peak torque may be critical for UCL and superior labral anterior posterior (SLAP) tears [14,16]. Similarly, strain on the UN is expected to increase during the MER. Strain of the UN increases with abduction and external rotation of the shoulder and flexion of the elbow [17]. Therefore, the mechanism of medial elbow pain during pitching should be investigated in each phase of pitching, because each phase of pitching applies different mechanical stress to structures supporting the medial elbow.

We aimed to clarify changes cross-sectional area (CSA) in flexor pronator muscles and the ulnar nerve (UN) in young subjects with medial elbow pain during pitching using ultrasonography and to investigate differences in those structure between pitching phases. We focused on the relationship between the flexor pronator muscles and the UN, and hypothesized that there may be US findings such as the flexor pronator muscles and UN changes young baseball players with medial elbow pain.

## 2. Methods

### 2.1. Design

The design of this study was conducted in a cross-sectional study. We recruited baseball players between June 2019 and March 2021 to have ultrasound measurements on both elbows.

### 2.2. Sample

Forty-two male junior high school, high school, and university baseball players with and without medial elbow pain during throwing that were recruited from one orthopedic clinic participated in this study. The number of participants included 20 junior high school players, 17 high school players, and 5 university players. Subjects were explained the summary and purpose of the study verbally and their signed consent was obtained. Informed consent was obtained from all subjects involved in the study. This study was approved by the Ethics Committee of our institution (authorization number, 2019-019). The medial elbow pain group consisted of 26 male baseball players and the Healthy group consisted of 16 male baseball players. The inclusion criterium of medial elbow pain group was players who had medial elbow pain during pitching, positive ulnar nerve neurodynamic test, Tinel’s sign and motor or sensory ulnar neuropathy symptoms. Ulnar nerve neurodynamic test was tested in shoulder abduction, maximum elbow flexion, forearm rotation, and maximum dorsiflexion of the wrist. Motor ulnar neuropathy was defined as weakness of the abductor and opponens digiti minimi muscles. Sensory ulnar neuropathy was defined as numbness or paresthesia in the fifth digit and ulnar halves of the fourth digit. After, the players were divided into MER and BR groups according to the pitching phase in which medial elbow pain occurred. The pitching phases in which pain occurs were interviewed by the subjects. In the medial elbow pain group, players were excluded who showed osteochondritis dissecans or olecranon disorder or who did not have pain during throwing. The inclusion criterium of healthy group was players who did not medial elbow pain during pitching and ulnar neuropathy symptoms in the past or present.

### 2.3. Instruments

The medial elbow was imaged by B mode ultrasonography (Noblus, Hitachi Aloca, Tokyo, Japan) with an 18-MHz linear-array transducer. Measurement was performed with the subject in the sitting position, with elbow flexion at 45° and the forearm supinated with the elbow to forearm placed on the upper limb table (Figure 1).

### 2.4. Procedures

The imaged region was a position 20% proximal from the medial epicondyle of the humerus to the ulnar styloid process, and the flexor digital profundus (FDP), flexor carpi ulnaris (FCU), flexor digitorum superficialis (FDS), and pronator teres (PT) muscles were imaged. This position is where the CSA of the target muscle can be measured. The method of identifying each FPM involved performing an active movement specific to each FPM, identifying the target muscle, and determining the imaging portion (Figure 2). Each muscles was exercised active, and pre-practice exercises were performed before measurement. First, the CSA at rest was imaged, and then each FPM was automatically moved to image the CSA during contraction (Figure 3). In the image evaluation, the CSA of each muscle at rest and at contraction were measured using the tracing function in the ultrasonography software. In addition, the UN was identified deep in the FCU, and the CSA of the UN at rest was measured (Figure 4). One examiner performed the procedure from imaging to measurement.

#### 2.4.1. Calculation Method

The muscle cross-sectional area was calculated as the CSA of the throwing side relative to the non-throwing side as follows. Because, the reason is that each subject has a different muscle CSA. The cross-sectional area of the UN was also calculated using the same formula.
$$\mathrm{Muscle}\;\mathrm{or}\;\mathrm{UN}\;\mathrm{CSA}=\frac{throwing\;CSA}{non\;throwing\;CSA}\times 100$$

#### 2.4.2. Intra-Rater Reliability and Power Analysis

The intra-rater reliability of this analysis method was examined in advance. One examiner measured 9 healthy adults according to the above method. The intra-rater reliability defined as poor (<0.5), moderate (0.5–0.75), good (0.75–0.9), and excellent (>0.9), and as absolute reliability, the standard error of measurement (SEM) was calculated. The number of samples required for the experiment was analyzed using G-power.

#### 2.4.3. Comparison between the Three Groups

All data are presented as means (standard deviations) and CSA ratio (per-centages) for continuous. The CSA of each muscles at rest and contraction and the CSA of UN in the healthy group, the MER group, and the BR group was compared among the 3 groups using one-way analysis of variance. After that, Tukey–Kramer test was performed as a multiple comparison test after one-way analysis of variance. In addition, 95% confidence intervals and effect sizes were calculated. A *p*-value < 0.05 was considered significant. Statistical analyses were carried out by the same examiner using SPSS Statistics ver.25.

#### 2.4.4. Correlation Analysis

Moreover, in order to clarify the relationship between the flexor pronator muscles at rest and contraction and UN, a single correlation analysis was performed. A *p*-value < 0.05 was considered significant. Statistical analyses were carried out by the same examiner using SPSS Statistics ver.25.

#### 2.4.5. ROC Analysis

We clarified the cutoff value for detecting medial elbow pain with ulnar neuropathy by phase of pitching. The cut-off values of the CSA of the UN and the CSA ratio of the UN (CSA of the throwing arm divided by that of the non-throwing arm) were determined for medial elbow pain by receiver operating characteristic (ROC) analysis. Sensitivity and specificity of each potential predictor of medial elbow pain were calculated, along with the area under the ROC curve (AUC) and its 95% confidence interval (95% CI). A *p*-value < 0.05 was considered significant. Statistical analyses were carried out by the same examiner using SPSS Statistics ver.25.

## 3. Results

### 3.1. Intra-Rater Reliability and Power Analysis

Intra-rater reliability was highly reproducible (Table 1). As a result of calculating the required sample size using the test force analysis software G-Power at the start of the experiment, the number of samples per group was calculated to be 13.

### 3.2. Baseball Player Demographics

The average age of the subjects was 15.3 years (Table 2). As a result, 13 players had pain during the MER and 13 players had pain during the BR. In the MER group, 8 players were pitchers and 5 were fielders. In the BR group, 5 players were pitchers and 8 were fielders. Four subjects were pitchers and 12 were fielders in the healthy group (Table 1). All participants have at least 3 years of baseball experience. There were no significant differences in exercise amount (times/week), and exercise time (number of hours) between the 3 groups (Table 2). The clinical diagnosis of ulnar neuropathy is shown in Table 3.

### 3.3. Comparison between the Three Groups

In comparison of the flexor pronator muscles’ CSA among the 3 groups at rest, there was a significant difference only in the FCU between the healthy group and the MER group (95.4 ± 15.5% vs. 76.6 ± 12.5%, *p* = 0.004) (Table 4). The 95% confidence intervals and effect sizes for the healthy and MER groups were 5.7–31.0 and 1.32, respectively.

On contraction, there was a significant difference only in the FCU between the healthy group and MER group (97.6 ± 12.8% vs. 84.8 ± 13.7%, *p* = 0.046) (Table 5). The 95% confidence intervals and effect sizes for the healthy and MER groups were 0.2–24.5 and 0.94, respectively.

There were significant differences in the CSA of the UN between the healthy and MER groups (105.0 ± 27.7% vs. 176.4.8 ± 53.5%, *p* = 0.001), and between the MER and BR groups (176.4.8 ± 53.5% vs. 132.9 ± 21.1%, *p* = 0.038). The 95% confidence intervals and effect sizes for the healthy and MER groups were 29.3–110.5 and 1.69, respectively. The healthy and MER groups were 2.2–81.9 and 1.06, respectively. There was also the significant difference between the healthy group and BR group (105.0 ± 27.7% vs. 132.9 ± 21.1%, *p* = 0.013) (Table 4). The 95% confidence intervals and effect sizes for the healthy and MER groups were 5.4–50.3 and 1.13, respectively.

### 3.4. Correlation

A negative correlation between each FPM and the UN was observed only in the FCU at rest (*p* = 0.027) (Table 6). There was no correlation between other muscles and the UN.

### 3.5. ROC Analysis

The ROC curve illustrating the diagnostic accuracy of the measured CSA of the UN and the CSA ratio of the UN in the detection of medial elbow pain is presented. The value for detecting medial elbow pain by the CSA of the UN exhibited a higher AUC in the CSA ratio than in the measured CSA (CSA:0.80, CSA ratio:0.95) (Figure 5). Sensitivity, specificity, and cut-off values were 100%, 88%, and 127%, respectively, for detecting medial elbow pain according to the CSA ratio.

## 4. Discussion

In this study, we divided the medial elbow pain group into the MER group and BR group based on the pitching phase with medial elbow pain, and we compared the CSA of both the flexor pronator muscles and UN among these two groups and the healthy group. As a result, we identified two structural changes, atrophy of the FCU and swelling of the UN, based on the pitching phase in which medial elbow pain was experienced. Moreover, there was a negative correlation between the FCU and UN at rest. These results suggest that the FCU atrophies as the UN hypertrophies. In addition, ROC analysis of the measured value of CSA and CSA ratio, the CSA ratio showed a higher AUC. These results suggest that UN can be quantitatively assessed by the CSA ratio, and that the CSA ratio greater than 127% may result in medial elbow pain.

During pitching, especially during the MER of pitching, valgus stress is applied to the medial elbow. As this stress continues to be applied to the medial elbow, repetitive mechanical stress on the UN increases [18]. The UN is known to be compressed at various positions in the elbow [10]. The measurement position in this study was distal to the cubital tunnel outlet, with potential compression of the aponeurotic attachments of the two heads of the FCU muscle. There are 3 layers of connective tissue (fascia) on the surface of the UN, and repeated throwing causes inflammation and degeneration of this connective tissue, which restricts nerve gliding and leads to nerve compression [19]. Chang et al. reported that nerve CSA is the most commonly used measurement for the diagnosis of entrapment neuropathies because the involved nerve tends to flatten at the compressive site and becomes swollen proximal to the level of compression [20]. In addition, Tajika et al. reported that FCU muscle strength correlates with pitching performance score and grip strength, and the importance of the FCU has been clarified in previous studies [16]. The results of this study also revealed atrophy of the FCU in the MER group, however, the cause of the atrophy is not clear. The athletes with medial elbow pain have findings of ulnar neuropathy, suggesting that FCU atrophy and UN may be involved in medial elbow pain [18]. However, the relationship between the FCU and UN was not clarified. The results of this study showed a negative correlation between the CSA of both the FCU and the UN, which suggests that athletes with pain during the MER had FCU atrophy in addition to ulnar neuropathy. Therefore, ulnar neuropathy due to compression and strangulation may have caused atrophy of the FCU, which is innervated by the UN. Additional testing, such as nerve conduction studies, should be conducted to evaluate the relationship between UN and FCU.

In the MER of pitching, the elbow flexes approximately 90–120° [21]. According to a report by Aoki et al., when the elbow joint flexes more than 120°, the nerve strain rate exceeds 13% [22]. It has been reported that the elasticity limit of the nerve due to traction is a strain rate of 15%, so stress near the elastic limit is applied to the UN at each pitch. Athletes with elbow valgus instability due to UCL injury are subject to greater extensional stress on the UN [23]. Mihata et al. reported that UN length and tension increased with increasing elbow valgus instability [17]. Repeated throwing may increase the risk of ulnar neuropathy due to excessive stretching stress on UN. When a peripheral nerve is inflamed due to repeated stresses, even 3% of nerve elongation can provoke pain and other nerve symptoms [24]. Therefore, our results suggest that athletes with pain in the injury group, especially those in the MER groups, may have pain during pitching due to complications of ulnar neuropathy.

Pain of ball release may be related to the flexor pronator muscles. Flexor pronator muscle injury near the origin of the medial epicondyle was observed at the time of operative treatment for 13% of pitchers with UCL failure in a large case series, highlighting the possibility of concomitant injury to the flexor-pronator mass among pitchers with valgus instability [6]. It is expected that excessive load is applied to the flexor pronator muscles due to the eccentric contraction of the flexor pronator muscles at the time of release. Since the flexor pronator muscles are dynamic stabilization for elbow valgus movement, excessive valgus stress is applied when muscle injury of the flexor pronator muscles accompanies UCL injury. Therefore, it is possible that the UN of the athletes in the BR group was more swollen than that of the healthy group. In addition, the median nerve penetrates between both PT heads [25,26]. Compression or entrapment of the both PT heads may cause median neuropathy. However, this was not addressed in the present study and needs to be clarified in the future.

In the past decade, the incidence of UCL injury has increased each year, and the cause of medial elbow pain is considered to be due to UCL injury [27,28]. However, 61% of asymptomatic athletes have abnormal findings of the UCL [29]. Thus, the relationship between abnormal findings of UCL and has not been clarified completely. In other words, in addition to the UCL, the flexor pronator muscles and the UN are important in elbow throwing injuries. Measurement of CSA is used commonly to evaluate ulnar neuropathy. Ulnar neuropathy is suggested to include >10 mm square of the CSA of the UN [20]. However, the CSA of a nerve is reported to correlate with subject height. [30] The subjects of this study ranged from junior high school students to university students, and their body sizes varied greatly. Therefore, it is necessary to create a different cutoff value dependent on subject age and height. In this study, ROC analysis of the measured value of CSA and CSA ratio, the CSA ratio showed a higher AUC. These results suggest that the flexor pronator muscles and the UN can be quantitatively evaluated by the CSA ratio.

There are several limitations of this study. The limitations of this study are that the pain area of the athlete is different from that of the measurement area, that it is a cross-sectional study, there is no longitudinal study, and the degree of muscle contraction was not uniform. For this study, US measurement of CSA was performed at one location, distal to the cubital tunnel outlet, and other regions where entrapment can be expected were not considered. Furthermore, as a cross-sectional study, it was not possible to investigate how pain changed with subsequent changes of the UN, so a prospective study is needed. In addition, UCL integrity was not described, which is a confounding variable in this study and a potential cause of medial elbow pain, and small sample size from a clinical standpoint. In the future, verification with a larger sample size may clarify the relationship with other flexor pronator muscles. Moreover, it is considered that a limitation of this study is that the relationship between CSA of the UN and the physical findings of the nerve has not been demonstrated. In the future, the relationship between medial elbow pain and the UN may be further clarified by increasing the number of UN measurement areas and prospectively observing changes in the UN. The relationship with other neuropathies, such as median neuropathy, also needs to be clarified.

## 5. Conclusions

Our results showed that the resting FCU of the MER group was atrophied and the UN was enlarged. In addition, a negative correlation was observed between the FCU and UN during rest and contraction. Our results suggest that athletes with medial elbow pain during the MER of pitching have ulnar nerve swelling. Therefore, it is necessary to evaluate the UN for rehabilitation of athletes with medial elbow pain during the MER of baseball pitching.

## Figures and Tables

**Figure 1 healthcare-11-00050-f001:**
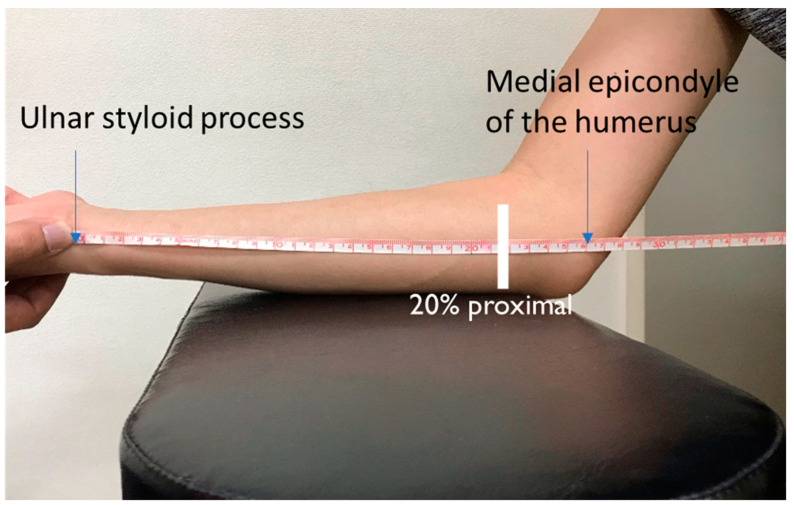
Clinical images at rest and contraction in the pitching arm.

**Figure 2 healthcare-11-00050-f002:**
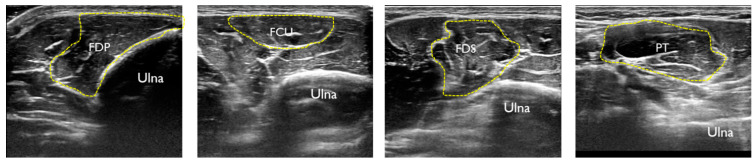
US cross-sectional image of each the flexor pronator muscles at the proximal 20% of the forearm.

**Figure 3 healthcare-11-00050-f003:**
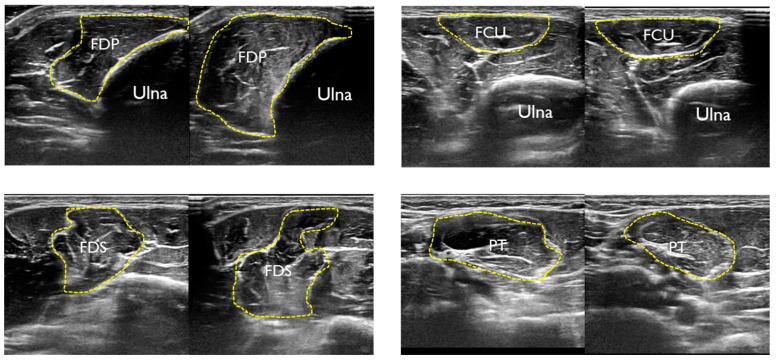
US images of each the flexor pronator muscles at rest and during contraction.

**Figure 4 healthcare-11-00050-f004:**
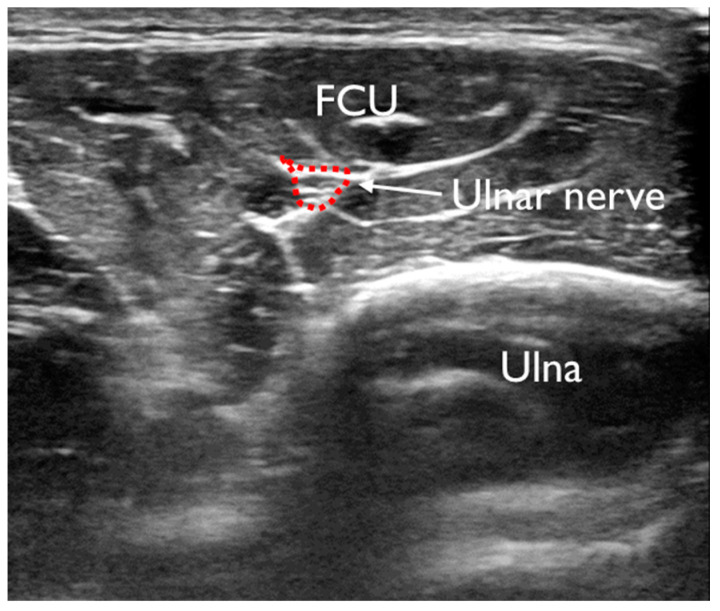
US image of ulnar nerve2.5. Statistics.

**Figure 5 healthcare-11-00050-f005:**
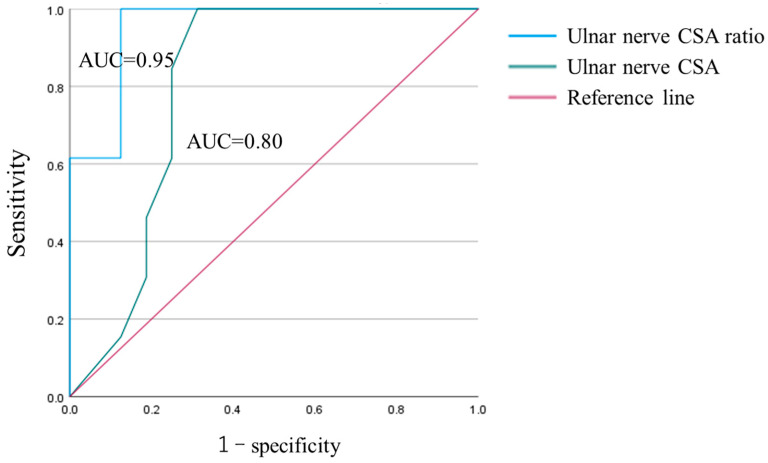
ROC curve.

**Table 1 healthcare-11-00050-t001:** Intra-rater reliability (ICC1.2).

	Rest			Contraction		
	ICC	95%CI	SEM	ICC	95%CI	SEM
FDP	0.93	0.69–0.98	0.17	0.89	0.56–0.98	0.20
FCU	0.99	0.94–0.99	0.05	0.98	0.92–0.99	0.07
FDS	0.94	0.76–0.99	0.19	0.94	0.77–0.99	0.13
PT	0.96	0.82–0.99	0.19	0.97	0.87–0.99	0.15
UN	0.90	0.56–0.98	0.005			

FDP: flexor digital profundus; FCU: flexor carpi ulnaris; FDS: flexor digitorum superficialis; PT: pronator teres; UN: ulnar nerve.

**Table 2 healthcare-11-00050-t002:** Baseball player demographics.

	Healthy Group (*n* = 16)	MER Group (*n* = 13)	BR Group (*n* = 13)
Age	14.3 ± 1.7	16.4 ± 2.2	15.8 ± 2.9
Height (cm)	164.7 ± 8.5	171.0 ± 10.2	165.6 ± 15.6
Weight (kg)	56.1 ± 10.6	64.9 ± 13.1	59.6 ± 16.3
Exercise amount (times/week)	3.9 ± 1.9	5.7 ± 1.4	4.1 ± 1.8
Exercise time (number/hours)	6.3 ± 1.9	4.1 ± 1.6	5.6 ± 2.7
Position	Pitcher 4 Fielder 12	Pitcher 8 Fielder 5	Pitcher 5 Fielder 8

**Table 3 healthcare-11-00050-t003:** Clinical diagnosis for ulnar neuropathy.

	MER Group (*n* = 13)	BR Group (*n* = 13)
Positive ulnar nerve neurodynamic test	13	9
Positive Tinel’s sign	10	5
Motor ulnar neuropathy	13	6
Sensory ulnar neuropathy	6	2

**Table 4 healthcare-11-00050-t004:** Comparison of rest CSA between the three groups.

	Group	Average ± SD	95%CI	One-Way ANOVA	Effect Size	Post Hoc
FDP	healthy	106.8 ± 45.3	82.6–131.0	*p* = 0.36	vs. MER *p* = 0.22, vs. BR *p* = 0.26	vs. MER *p* = 0.81, vs. BR *p* = 0.97
	MER	100.0 ± 21.1	86.3–111.3		vs. BR *p* = 0.09	vs. BR *p* = 0.74
	BR	96.7 ± 28.7	79.3–114.0			
FCU	healthy	95.4 ± 15.5	87.1–103.6	*p* = 0.01	vs. MER *p* = 1.30, vs. BR *p* = 0.40	vs. MER *p* = 0.003 **, vs. BR *p* = 0.55
	MER	76.6 ± 12.5	69.7–84.3		vs. BR *p* = 0.78	vs. BR *p* = 0.14
	BR	88.7 ± 17.6	78.1–99.4			
FDS	healthy	102.8 ± 28.6	87.5–118.1	*p* = 0.82	vs. MER *p* = 0.04, vs. BR *p* = 0.23	vs MER *p* = 0.99, vs BR *p* = 0.82
	MER	105.7 ± 36.4	82.7–125.4		vs. BR *p* = 0.18	vs BR *p* = 0.90
	BR	110.4 ± 36.8	88.1–132.6			
PT	healthy	128.5 ± 32.6	111.1–145.9	*p* = 0.85	vs. MER *p* = 0.16, vs. BR *p* = 0.20	vs. MER *p* = 0.90, vs. BR *p* = 0.85
	MER	120.8 ± 41.2	98.4–146.8		vs. BR *p* = 0.01	vs BR *p* = 1.00
	BR	122.8 ± 23.0	109.0–136.7			
UN	healthy	105.0 ± 27.7	90.3–119.8	*p* = 0.00002	vs. MER *p* = 1.75, vs. BR *p* = 1.11	vs. MER *p* = 0.001 **, vs. BR *p* = 0.01 **
	MER	176.4 ± 53.5	143.8–206.1		vs. BR *p* = 1.07	vs. BR *p* = 0.04 *
	BR	132.9 ± 21.1	120.1–145.6			

FDP: flexor digital profundus; FCU: flexor carpi ulnaris; FDS: flexor digitorum superficialis; PT: pronator teres; UN: ulnar nerve. ** *p* < 0.01; * *p* < 0.05

**Table 5 healthcare-11-00050-t005:** Comparison of contraction CSA between the three groups.

	Group	Average ± SD	95%CI	One-Way ANOVA	Effect Size	Post Hoc
FDP	healthy	109.8 ± 30.2	93.7–125.9	*p* = 0.19	vs. MER *p* = 0.70, vs. BR *p* = 0.38	vs. MER *p* = 0.14, vs. BR *p* = 0.56
	MER	94.7 ± 16.1	81.1–102.9		vs. BR *p* = 0.29	vs. BR *p* = 0.74
	BR	98.8 ± 26.9	98.8 ± 26.9			
FCU	healthy	97.6 ± 12.8	90.8–104.5	*p* = 0.04	vs. MER *p* = 0.95, vs. BR *p* = 0.19	vs. MER *p* = 0.04 *, vs. BR *p* = 0.87
	MER	84.8 ± 13.7	77.3–93.3		vs. BR *p* = 0.66	vs. BR *p* = 0.23
	BR	94.9 ± 15.8	94.9 ± 15.8			
FDS	healthy	102.9 ± 30.3	86.8–119.1	*p* = 0.95	vs. MER *p* = 0.12, vs. BR *p* = 0.06	vs. MER *p* = 0.95, vs. BR *p* = 0.99
	MER	112.1 ± 33.3	84.6–129.3		vs. BR *p* = 0.06	vs. BR *p* = 0.99
	BR	104.9 ± 36.2	104.9 ± 36.2			
PT	healthy	132.2 ± 42.7	109.4–154.9	*p* = 0.29	vs. MER *p* = 0.47, vs. BR *p* = 0.15	vs. MER *p* = 0.41, vs. BR *p* = 0.92
	MER	111.2 ± 35.0	92.7–134.5		vs. BR *p* = 0.60	vs. BR *p* = 0.29
	BR	138.9 ± 48.0	138.9 ± 48.0			

FDP: flexor digital profundus; FCU: flexor carpi ulnaris; FDS: flexor digitorum superficialis; PT: pronator teres; UN: ulnar nerve. * *p* < 0.05

**Table 6 healthcare-11-00050-t006:** Correlation between three groups of the flexor pronator muscles and UN.

	Rest				Contraction			
	FDP	FCU	FDS	PT	FDP	FCU	FDS	PT
UN	−0.20	−0.34 *	0.26	−0.06	−0.20	−0.17	0.26	−0.10

FDP: flexor digital profundus; FCU: flexor carpi ulnaris; FDS: flexor digitorum superficialis; PT: pronator teres; UN: ulnar nerve. * *p* = 0.027.

## Data Availability

The data are available upon request from the corresponding author.

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
