# Peer review of "Relationship between Medial Elbow Pain, Flexor Pronator Muscles, and the Ulnar Nerve in Baseball Players Using Ultrasonography"

_healthcare, 2022, doi:10.3390/healthcare11010050_

Round 1
Reviewer 1 Report
First of all, I would like to congratulate the authors of the research idea. The work contains many interesting issues and draws attention to an important and common problem. However medial elbow pain does not affect only young athletes.
Below are the following notes:
1. The title suggests that there may be a connection with the whole group of flexor pronator muscles - the flexor digital profundus (FDP), flexor carpi ulnaris (FCU), flexor digital superficialis (FDS) - and the ulnar nerve. The anatomical course of the ulnar nerve and clinical practice show that the ulnar nerve neuropathy may have an obvious relationship with FCU. One might possibly expect a connection with the FDP, but not with the FDS or the PT. Considering the above, it seems that the authors did unnecessary work evaluating the PDS or PT, which does not diminish the value of their work. However, in the case of PT, the authors should at least mention the possibility of median nerve neuropathy due to irritation at the point of penetration between both PT heads.
2. At least the other most common pathologies causing medial elbow pain should have been mentioned. Perhaps they are rarer at such a young age as the average age of the study population. I consider it necessary to list them, especially since the authors use ultrasonography. Ultrasonography plays an important role in the diagnosis of these pathologies. See at least https://doi.org/10.3390/healthcare10081529.
3. In the introduction, the authors describe the forces acting on the UCL. The UCL is the most important passive stabilizer of the elbow joint under valgus forces. The second is the head of the radius. It is obvious that the UCL does not break when its maximum strength is exceeded when pitching, because active stabilizers also work then, but the head of the radius also resists.
4. It is noteworthy that the study group and the control group are small, but the statistics are compiled correctly.
5. The tests shown in Figure 3. are not specific. They involve not only the described muscles, but also others. Performing the test like in image a. (FDP) is practically impossible. I propose to remove completely Figure 3.
6. Please explain why electromyography was not performed when clinical signs of ulnar neuropathy were found.
7. Please explain why the control (healthy) group was not clinically tested for ulnar neuropathy. Was it assumed that the control group could not have symptoms of neuropathy. Is the clinical examination of these symptoms really so highly specific to ulnar neuropathy?
8. While reading, many other doubts arise, which, fortunately, the authors dispel in the discussion. The discussion is a very good part of the article.
9. Authors should include references to more recent publications.
Reviewer 2 Report
Manuscript shows the relation between medial elbow pain, flexor pronator muscles and ulnar nerve in baseball players using ultrasonography. I have some considerations for you.
Line 97-104 are results.
Table 1 and table 2 should be moved to results section.
Methods section are quite confusing. There are a lot of continuous information without order. Maybe you could add subsections as participants, procedure, variables…
Although you have explained it, the three different groups do not seem clear. Why did you decide this classification?
Figure 2 and 4 are results too.
Table 3b and 3c are figures.
The variety of abbreviations makes sometime difficult to follow the reading.
Line 215: add a full estop after “al”.
Reference: please revise MDPI rules about reference format.
Writing must be reviewed. Full stops should go after the reference.
Reviewer 3 Report
As per notes attached

Round 2
Reviewer 1 Report
The authors' corrections are sufficient.
Reviewer 2 Report
Dear authors,
Thank you for considering my suggestions.
You have resolved my doubts.
I have one point to add: Line 222: table explanation should be before the table like the others.
Reviewer 3 Report
In view of the adaptations presented, I consider the manuscript ready to be published.